# Preconditioned cues have no value

**Melissa J Sharpe[1,2,3]\*, Hannah M Batchelor[1], Geoffrey Schoenbaum[1,4,5,7]\***

[1]NIDA Intramural Research Program, Baltimore, United States; [2]Princeton Neuroscience Institute, Princeton University, Princeton, United States; [3]School of Psychology, University of New South Wales, Sydney, Australia; [4]Department of Anatomy and Neurobiology, University of Maryland School of Medicine, Baltimore, United States; [5]Department of Psychiatry, University of Maryland School of Medicine, Baltimore, United States; [7]Solomon H. Snyder Department of Neuroscience , The Johns Hopkins University, Baltimore, United States

**Abstract** Sensory preconditioning has been used to implicate midbrain dopamine in model-based learning, contradicting the view that dopamine transients reflect model-free value. However, it has been suggested that model-free value might accrue directly to the preconditioned cue through mediated learning. Here, building on previous work (Sadacca et al., 2016), we address this question by testing whether a preconditioned cue will support conditioned reinforcement in rats. We found that while both directly conditioned and second-order conditioned cues supported robust conditioned reinforcement, a preconditioned cue did not. These data show that the preconditioned cue in our procedure does not directly accrue model-free value and further suggest that the cue may not necessarily access value even indirectly in a model-based manner. If so, then phasic response of dopamine neurons to cues in this setting cannot be described as signaling errors in predicting value.

DOI: https://doi.org/10.7554/eLife.28362.001

**\*For correspondence:**
melissa.sharpe@nih.gov (MJS);
geoffrey.schoenbaum@nih.gov
(GS)

## Introduction

Behaviour is often divided into two broad categories. One, termed goal-directed or model-based, utilizes an associative map of the task at hand, which can be navigated to anticipate likely outcomes and their desirability. Maps acquired separately can be linked and the value of outcomes updated on-the-fly to allow flexible responding. The other, contrasting category of behaviour, termed model-free or habitual, reflects simpler associations linking cues to the responses that have been reinforced in their presence. Behaviours in both categories are typically described as reflecting value, however in the former category, the value is inferred and reflects value stored downstream, whereas in the latter, the value is directly attached or 'cached' in the antecedent cue.

Our lab has recently used sensory preconditioning to identify neural systems critical for model-based behaviour (*Jones et al., 2012*; *Wied et al., 2013*; *Sadacca et al., 2016*; *Sharpe et al., 2017*). These data include the demonstration that midbrain dopamine neurons exhibit error-like activity to preconditioned cues. Our use of this task is based on the belief that the design is particularly effective in isolating model-based behaviour from behaviour reflecting model-free value. In sensory preconditioning, two neutral cues are paired together in close succession such that a relationship can form between them (e.g. A→B). While there are no observable changes to behaviour during this phase, the existence of this association can be revealed if cue B is paired with reward, which causes subjects to start responding to A as if they expect reward to be delivered. Indeed, responding to cue A is sensitive to the current desire for the food reward at the time of the probe test (*Blundell et al., 2003*). From data such as these, it is thought that subjects respond to the preconditioned cue either because A evokes a representation of B and B leads to thoughts of reward during the test phase, or because B evokes a representation of A during conditioning that allows A to

become directly associated with reward (*Jones et al., 2012*; *Wimmer and Shohamy, 2012*; *Gershman, 2017*). Thus, sensory preconditioning seems to be an iconic example of a model-based behaviour.

However, while it is clear that sensory preconditioning utilizes model-based associations, this procedure may also permit the preconditioned cue to directly accrue value. Specifically, if presentation of cue B were to evoke a representation of cue A during conditioning, then the value of the food might become directly associated with A (*Wimmer and Shohamy, 2012*; *Doll and Daw, 2016*). Importantly, this question is not resolved by the effect of food devaluation on responding to the preconditioned cue, since the cue could maintain any such model-free value subsequently, independent of the new value of the food as the association between cue A and the devalued food has not been directly experienced. If this were occurring in our procedure, it would introduce difficulties in its use to strictly isolate model-based neural processing. For example, the ability of a preconditioned cue to evoke phasic activity in a dopamine neuron could be easily explained by existing proposals that dopaminergic transients reflect errors in predicting model-free value (*Schultz et al., 1997*).

Here we directly addressed this question by assessing the ability of a preconditioned cue trained in our task to support conditioned reinforcement. For comparison, we also assessed conditioned reinforcement supported by cues trained to predict reward directly or through second-order conditioning. Conditioned reinforcement – or the ability of a cue to support acquisition of an instrumental response in the absence of any reward - is generally conceptualised as a test of cue value. Notably, subjects will work for a cue predicting food even if the food reward has been devalued (*Parkinson et al., 2005*), indicating that model-free value is normally sufficient to support conditioned reinforcement. Accordingly, we found that both directly conditioned and second-order cues would support conditioned reinforcement. However, a preconditioned cue would not. These data show that, at least for our procedure in rats, the preconditioned cue does not acquire model-free value during training. Further they suggest that the cue also does not automatically or by default access value cached in events downstream in a model-based manner, such as through the other cue or the sensory properties of the reward.

## Results

### Preconditioned cues do not support conditioned reinforcement

#### Preconditioning
Rats were first presented with the neutral cues (A→B; C→D) in close succession 12 times each to promote the development of a relationship between them. As expected, since training did not involve presentation of reward, the rats spent little time in the magazine during this phase, and there were no differences between cues (*Figure 1A*). ANOVA revealed no main effect of cue ($F_{(3,63)}=2.12$, $p>0.05$).

#### Conditioning
Following preconditioning, rats underwent conditioning for 4 days. Each day, rats received 12 presentations of cue B followed by the delivery of two sucrose pellets (B→2US) and 12 presentation of cue D without reward (D→ no US). As training progressed, all rats acquired a conditioned response to cue B as indexed by a greater time spent in the magazine during presentation of this cue as training progressed (*Figure 1B*). A two-factor ANOVA (cue ×day) showed main effects of cue ($F_{(1,21)}=87.47$, $p<0.05$) and day ($F_{(3,63)}=4.45$, $p<0.05$) and an interaction between these factors ($F_{(3,63)}=21.42$, $p<0.05$).

#### Conditioned reinforcement tests
Following Pavlovian training, we next gave rats two conditioned reinforcement sessions. In the first test, pressing one lever led to a 2 s presentation of cue A (R1→A), and pressing the other lever led to a 2 s presentation of cue C (R2→C). Here, we found that rats made a small number of lever

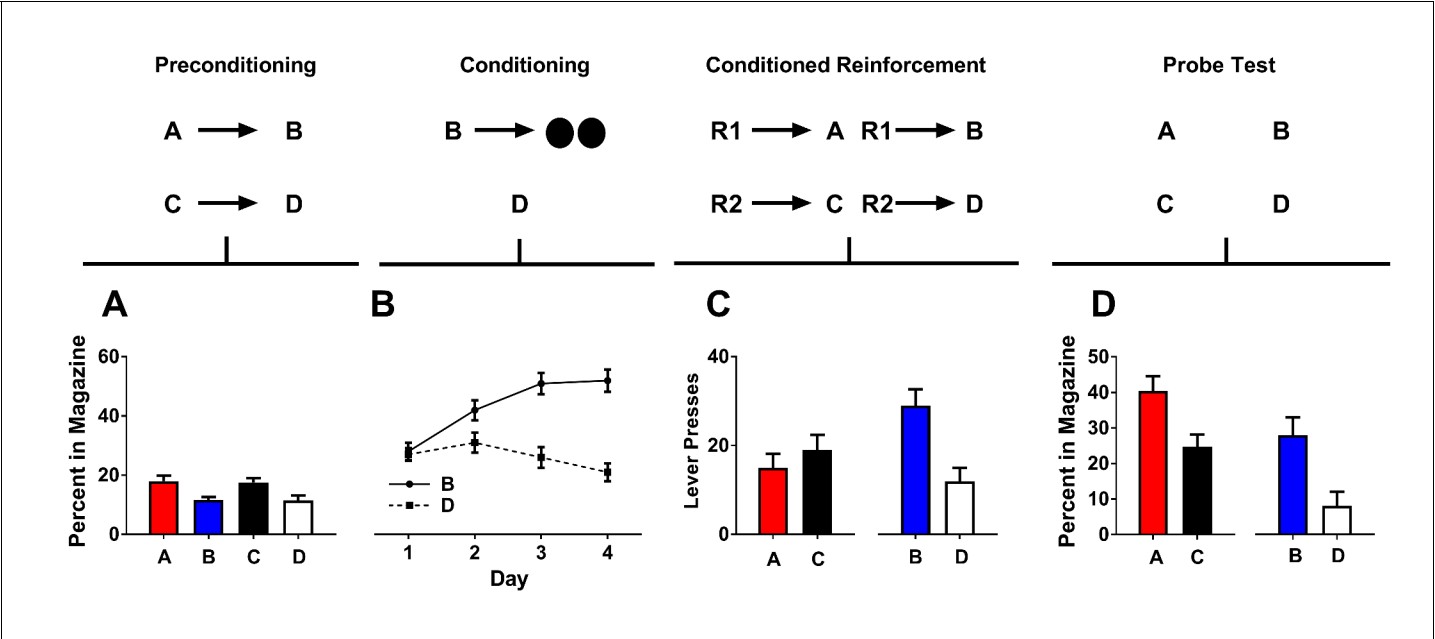

**Figure 1.** Preconditioned cues do not support conditioned reinforcement. Rates of responding are represented as percent time spent in the magazine during cue presentation (Figures A, B, and D) or number of lever presses (±SEM). Graphs show preconditioning (**A**), conditioning (**B**), conditioned reinforcement (**C**), and Pavlovian probe tests (**D**).

DOI: https://doi.org/10.7554/eLife.28362.002

presses on each lever and did not show any difference in the number of lever presses made for presentation of either cue (*Figure 1C*; left).

To ensure that we could obtain conditioned reinforcement in this cohort of rats, we gave rats another conditioned reinforcement test. In this test, one lever press led to a 2 s presentation of cue B (R1→ B) and the other lever press led to a 2 s presentation of cue D (R2→D). In contrast to the first conditioned reinforcement test, during this session rats showed a higher rate of lever pressing on the lever which produced the reward-paired cue B and a low level of lever presses for non-rewarded cue D (*Figure 1C*; right).

The difference in the pattern of results seen across the first and second session of the conditioned reinforcement tests was confirmed with statistical analyses. A two-factor ANOVA [cue type (preconditioned vs. conditioned)×reinforcement (rewarded or non-rewarded)] showed no effects of cue type (AC vs BD; $F_{(1,21)}=0.82$, $p>0.05$) or reinforcement (AB vs. CD; $F_{(1,21)}=1.44$, $p>0.05$), however there was a significant interaction between these factors ($F_{(1,21)}=10.92$, $p<0.05$). Simple-main effects analyses showed that the source of this interaction was due to a significant elevation in lever pressing for B that was not observed for the other cues (vs A: $F_{(1,21)}=7.64$, $p<0.05$; vs D: $F_{(1,21)}=7.38$, $p<0.05$; C vs. D: $F_{(1,21)}=3.08$, $p>0.05$; A vs. C: $F < 1$). Thus, preconditioned cues did not support conditioned reinforcement in the same rats that readily showed conditioned reinforcement for the cue directly paired with reward.

## Pavlovian probe tests

It is plausible that the reason we failed to see effective conditioned reinforcement with the preconditioned cue A was because rats failed to learn the relationship between A and B. In this case, they would be failing to press the lever because they were failing to generate the normal expectation, after conditioning, that A might lead to reward. In order to test this hypothesis, we next gave rats two Pavlovian probe tests to assess learning. In the first session, we gave rats unrewarded presentations of A and C; in the second session, we gave rats unrewarded presentations of B and D. We found that rats made more entries into the food port during presentation of either cue A or B, demonstrating effective conditioning and sensory preconditioning (*Figure 1D*). A two-factor ANOVA [cue type (preconditioned vs. conditioned)×reinforcement (rewarded or non-rewarded)] revealed a

main effect of reinforcement (AB vs CD; $F_{(1,21)}$=15.11, p<0.05). There was also a main effect of cue type (AC vs BD; $F_{(1,21)}$=9.39, p<0.05), likely reflecting that the A vs. C extinction tests were given prior to the B vs D tests since the A vs. C test is the critical comparison. Importantly, however, there was no interaction with cue type ($F < 1$). Thus, rats spent a greater amount of time in the food port during presentation of cues A and B relative to cues C and D, and there was no difference in the magnitude of this difference. In order to full rule out any possibility that the lack of conditioned reinforcement observed to the preconditioned cue A was due to a failure of sensory preconditioning, we also separately tested the difference between A and C. This analysis revealed a significant difference between responding to A and C ($F_{(1,21)}$=5.35, p<0.05).

## Second-order conditioned cues do support conditioned reinforcement

Our first experiment showed that a preconditioned cue is insufficient for conditioned reinforcement, whereas a cue directly paired with a valuable reward was sufficient. To confirm that this effect was not simply the result of the introduction of an additional cue between the preconditioned cue and the reward, we conducted a second experiment in which we tested the ability of a second-order conditioned cue to support conditioned reinforcement. Importantly, the second-order cue is trained exactly like the preconditioned cue except that the pairing of the neutral cues (A→B; C→D) occurs after rather than before training with reward (B→2US; D→ no US).

### Conditioning

Conditioning lasted for 4 days. Each day, the rats received 12 presentations of cue B followed by delivery of two sucrose pellets and 12 unrewarded presentations of cue D. As training progressed, all rats acquired a conditioned response to cue B (*Figure 2A*). A two-factor ANOVA (cue ×day) revealed a main effect of cue ($F_{(1,14)}$=37.13, p<0.05), a main effect of day ($F_{(1,14)}$=6.32, p<0.05), and a significant interaction between these factors ($F_{(1,14)}$=8.47, p<0.05).

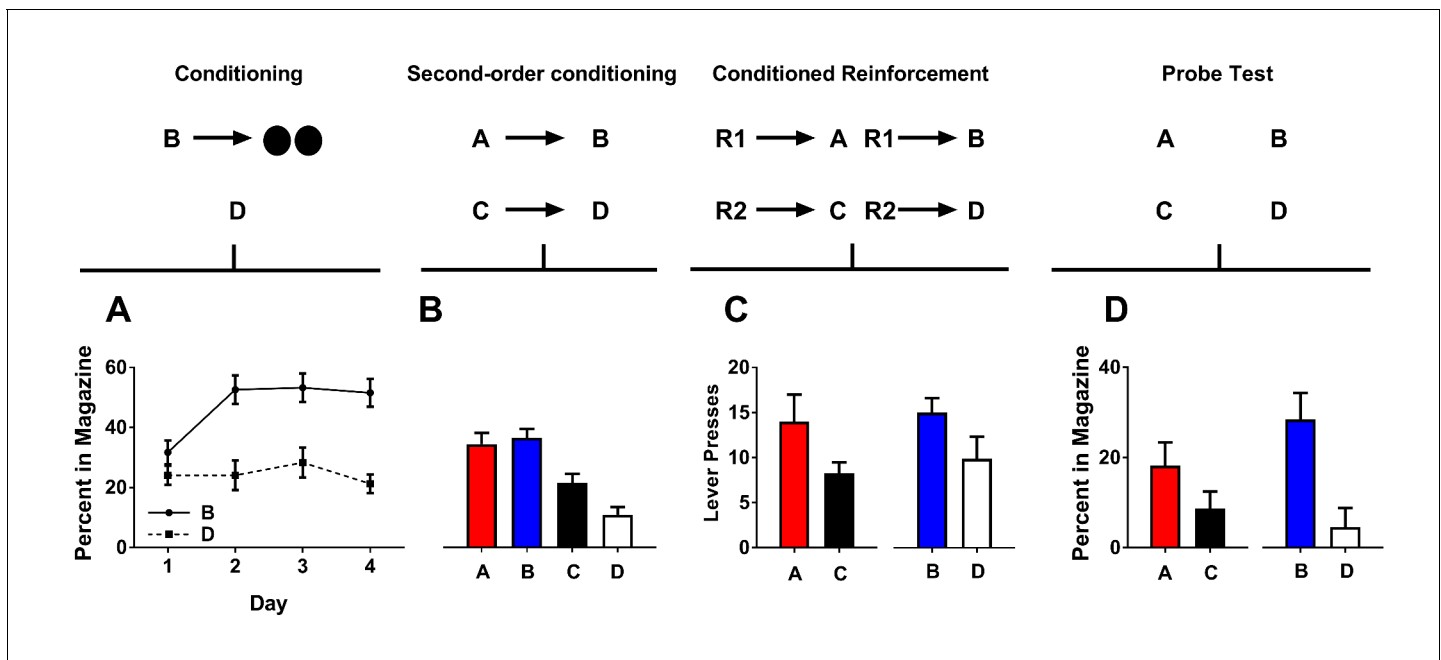

**Figure 2.** Second-order conditioned cues do support conditioned reinforcement. Rates of responding are represented as percent time spent in the magazine during cue presentation (Figures A, B, and D) or number of lever presses (±SEM). Graphs show preconditioning (**A**), conditioning (**B**), conditioned reinforcement (**C**), and Pavlovian probe tests (**D**).
DOI: https://doi.org/10.7554/eLife.28362.003

## Second-order conditioning

Following conditioning, rats were presented with the neutral cues (A→B; C→D) in close succession 12 times each to promote the development of a relationship between them. Rats spent more time in the magazine during cues A and B relative to cues C and D (*Figure 2B*). This was confirmed with statistical analyses. A two-factor ANOVA [cue type (second-order conditioned vs. conditioned)×reinforcement (rewarded or non-rewarded)] a main effect of reinforcement (AB vs CD; $F_{(1,14)}$=17.13, p<0.05), but no interaction ($F_{(1,14)}$=2.19, p>0.05) nor main effect of cue type (AC vs BD; $F_{(1,14)}$=4.04, p>0.05). Thus, rats spent a greater amount of time in the food port during presentation of cues A and B relative to cues C and D, and there was no difference in the magnitude of this difference.

## Conditioned reinforcement tests

Following second-order conditioning, we again gave rats two conditioned reinforcement tests. In the first, rats could press either lever for a 2 s presentation of cue A or C (R1→A; R2→C). In the second, rats could press these levers for either a 2 s presentation of cue B or D (R1→ B; R2→D). In both tests, we found that rats would press the lever more for the cue paired either directly or indirectly with reward (i.e. A and B relative to C and D; *Figure 2D*). A two-factor ANOVA [cue type (second-order conditioned vs. conditioned)×reinforcement (rewarded or non-rewarded)] showed a significant main effect of reinforcement (AB vs.CD; $F_{(1,14)}$=5.07, p<0.05), but no main effect nor any interaction with cue type (AC vs BD; $F < 1$). Thus A and B both supported conditioned reinforcement and did so to a similar degree.

## Pavlovian probe tests

Following the conditioned reinforcement tests, we gave rats two probe test to assess the ability of the cues A and B to promote entry into the food port. In the first, we gave rats unrewarded presentations of cue A and C. In the second, we gave rats unrewarded presentations of cue B and D. Rats spent a larger proportion of time in the magazine during presentation of cues A and B relative to cues C and D, confirming the second-order conditioning effect. A two-factor ANOVA [cue type (preconditioned vs. conditioned)×reinforcement (rewarded or non-rewarded)] revealed a main effect of reinforcement (AB vs CD; $F_{(1,14)}$=14.07, p<0.05) but no main effect nor any interaction with cue type (AC vs BD; $F < 1$). Thus, rats spent a greater amount of time in the food port during presentation of cues A and B relative to cues C and D and there was no difference in the magnitude of this difference.

## Discussion

Here we have shown that preconditioned cues do not support conditioned reinforcement. Rats showed no evidence of increased lever pressing for the cue trained to predict a cue that was later paired with reward. This was true despite strong responding at the food cup for the preconditioned cue in a subsequent probe test and robust conditioned reinforcement for the cue paired directly with food in the same rats. Further, in a second experiment, we also showed that a second-order cue supports conditioned reinforcement. Critically, our second-order conditioning procedures were identical to those used for sensory preconditioning, except for the order of training in second-order conditioning, which allowed the initial cue in the series to be paired with something of value at the time of conditioning.

In interpreting these data, it is important to emphasize that conditioned reinforcement is normally insensitive to devaluation of the food reward (*Parkinson et al., 2005*; *Burke et al., 2007*; *Burke et al., 2008*). In other words, if the food reward is devalued by pairing it with illness prior to conditioned reinforcement training, a cue that was previously paired with that reward will still support acquisition of lever pressing. Thus, value cached in the cue is normally sufficient to support the behaviour. Given this, our failure to detect any evidence of conditioned reinforcement for a preconditioned cue is strong evidence that a preconditioned cue does not accrue model-free value in this task.

This result has important implications for recent work using this task to investigate the neural circuits involved in model-based learning and behaviour (*Sadacca et al., 2016*; *Sharpe et al., 2017*). For example, we have recently shown that dopamine neurons exhibit phasic responses to both

directly- and pre-conditioned cues (*Sadacca et al., 2016*). We interpreted this result as showing that model-based information is reflected in dopaminergic error-signals, based on the presumption that the behaviour directed at the preconditioned cue is due to inference or model-based processing. This conclusion would be contrary to current proposals that these signals only reflect model-free value (*Sutton and Barto, 1981*; *Schultz et al., 1997*; *Schultz, 1998*; *Waelti et al., 2001*; *Schultz, 2002*; *Cohen et al., 2012*). However, it was proposed that the firing of the dopamine neurons to the preconditioned cue could reflect value that accrues to the cue via mediated learning in the conditioning phase or some other form of post-training rehearsal (*Doll and Daw, 2016*). The current results are inconsistent with this alternative interpretation. In particular, while our data do not rule out mediated learning as an underlying mechanism, they suggest that if responding to the preconditioned cue in our task is supported by mediated learning, as has been suggested in other designs and species (*Wimmer and Shohamy, 2012*), then that process does not cause the preconditioned cue to accrue model-free value.

Our data also raise questions as to whether preconditioned cues access, at least automatically or by default, any sort of stored value. As noted earlier, one way to think about responding to the preconditioned cue is as reflecting an inferred or model-based value. This is a value stored in downstream events and accessed through the associative model of the task acquired during prior training (*Jones et al., 2012*; *Wimmer and Shohamy, 2012*; *Gershman, 2017*). That is, in the probe test, the preconditioned cue evokes a representation of the sensory properties of the food reward, either directly or indirectly, and thereby activates the current value of the food. This view is consistent with the effects of devaluation, which normally eliminates responding to the food cup upon presentation of the preconditioned cue (*Blundell et al., 2003*). Yet if the preconditioned cue accesses the value stored in the food in this model-based manner, then one might have expected this cue to support conditioned reinforcement. This would make intuitive sense and is consistent with evidence that model-based value can support conditioned reinforcement (*Burke et al., 2007*; *Burke et al., 2008*). The failure of the preconditioned cue to support conditioned reinforcement suggests that it does not have automatic access to the value stored in the food, perhaps because it is never directly paired with anything that has value at the time. While speculative, this conclusion would have profound implications for interpreting the firing of dopamine neurons in this setting and perhaps in other tasks, where they exhibit phasic responses that are not obviously value based (*Horvitz, 2000*; *Tobler et al., 2003*; *Bromberg-Martin and Hikosaka, 2009*; *Sadacca et al., 2016*; *Takahashi et al., 2017*). These transient responses may signal the sensory, state, or informational error inherent in these designs, rather than anything related to a representation of value, model-based or otherwise.

## Materials and methods

### Subjects

Thirty-seven experimentally-naïve male Long-Evans rats (NIDA breeding program) were used in these experiments. Rats were maintained on a 12 hr light-dark cycle, where all behavioural experiments were conducted during the light cycle. Prior to behavioural testing, rats were placed on food restriction and maintained on ~85% of their free-feeding body weight. All experimental procedures were conducted in accordance with Institutional Animal Care and Use Committee of the US National Institute of Health guidelines.

### Apparatus, cues, and general procedures

Training was conducted in eight standard behavioural chambers (Coulbourn Instruments; Allentown, PA) individually housed in light- and sound-attenuating chambers. Each chamber was equipped with a pellet dispenser that delivered one 45 mg pellet into a recessed magazine when activated. Access to, and duration spent in, the magazine was detected by means of infrared detectors mounted across the mouth of the recess. The chambers contained an auditory stimulus generator, which delivered the tone and siren stimulus through a common speaker on the top right-hand side of the front chamber wall when activated. A second speaker on the back wall of the chamber, connected to another auditory stimulus generator, delivered the white noise stimulus. Finally, a heavy-duty relay delivering a 5 kHz clicker stimulus was located on the top left-hand side of the front chamber wall. During conditioned reinforcement tests, two levers were placed in the behavioural chamber, on the

left or right side of the front wall, and the magazine and pellet dispenser were removed. A computer equipped with Coulbourn Instruments software (Allentown, PA) controlled the equipment and recorded the responses. Cues A and C were either a white noise or clicker, and cues B and D were either a tone or siren (counterbalanced across rats). During Pavlovian training, stimuli were 10 s in length, and the order of trials was randomly intermixed and counterbalanced, with inter-trial intervals (ITI) averaging 6 min. During conditioned reinforcement testing, lever pressing produced 2 s of the relevant cue. Prior to training, all rats were shaped to enter the magazine to retrieve reward (two 45 mg sucrose pellets; 5TUT, Test Diet, MO), receiving 30 pellets in the magazine across a one hour period. Subsequently, rats received 2 sessions of training each day, one in the morning and one in the afternoon.

### Sensory preconditioning
Rats began with 2 sessions of compound cue training. In each session, rats received 6 presentations of serial compounds A→ B and C→ D, where cues A or C were immediately followed by presentation of cue B or D. Subsequently, rats underwent conditioning where cue B was followed by presentation of sucrose pellets while D was presented without reward. Rats received a total of 8 conditioning sessions with each consisting of six reinforced presentations of B and six non-reinforced presentation of D.

### Second-order conditioning
Rats began with 8 sessions of conditioning. In each session, rats received six reinforced presentations of B and six non-reinforced presentation of D. Subsequently, rats underwent 2 sessions of compound cue training, consisting of 6 presentations of serial compounds A→ B and C→ D, where cues A or C were immediately followed by presentation of cue B or D.

### Conditioned reinforcement and pavlovian probe tests
Following Pavlovian training, rats received two conditioned reinforcement tests each lasting 30 min. For these tests, levers were inserted in the chamber and the food magazine was removed (*Burke et al., 2008*). In the first test session, pressing one lever resulted in immediate 2 s presentation of cue A, while pressing the other lever resulted in a 2 s presentation of cue C (counterbalanced). In the second, the lever presses resulted in an immediate 2 s presentation of either cue B or D. To ensure that all animals learnt the associations promoted by sensory preconditioning, we also conducted two probe tests following conditioned reinforcement. In these tests, the levers were removed, and the food magazine was put back into the chamber. In the first probe test, rats received 6 presentation of cue A and C and magazine entries were measured. In the second, rats received six presentations each of cue B and D. No reward was presented during either the conditioned reinforcement or probe tests.

### Statistical analyses
Conditioned responding was measured as the fraction of time that the rats spent in the food magazine during cue presentation. This was restricted to the last five seconds when cues led to reward or a reward-paired cue, reflecting the normal escalation of responding towards the end of the cue when the reward is more likely to be delivered (i.e. inhibition of delay). Analyses on data from the final Pavlovian probe tests were conducted on the first two trials of each cue in the test session. Conditioned reinforcement was measured as the sum of the lever presses made across the full 30 min of each test session. All statistics were conducted using SPSS 24 IBM statistics package (*Sharpe and Killcross, 2014*). Generally, analyses were conducted using a mixed-design repeated-measures analysis of variance (ANOVA). All analyses of simple main effects were planned and orthogonal and therefore did not necessitate controlling for multiple comparisons.

## Acknowledgements
This work was supported by the Intramural Research Program at the National Institute on Drug Abuse (ZIA-DA000587) and a CJ Martin overseas biomedical fellowship awarded to MJS (National

Health and Medical Research Council, Australia). The opinions expressed in this article are the authors' own and do not reflect the view of the NIH/DHHS.

## Additional information

### Competing interests
Geoffrey Schoenbaum: Reviewing editor, *eLife*. The other authors declare that no competing interests exist.

### Funding

| Funder | Grant reference number | Author |
|---|---|---|
| National Health and Medical Research Council | APP1122980 | Melissa J Sharpe |
| National Institute on Drug Abuse | Intramural Research Program zia-da000587 | Geoffrey Schoenbaum |

The funders had no role in study design, data collection and interpretation, or the decision to submit the work for publication.

### Author contributions
Melissa J Sharpe, Conceptualization, Formal analysis, Investigation, Visualization, Methodology, Writing—original draft, Project administration, Writing—review and editing; Hannah M Batchelor, Investigation, Writing—review and editing; Geoffrey Schoenbaum, Conceptualization, Resources, Supervision, Funding acquisition, Investigation, Methodology, Writing—original draft, Project administration, Writing—review and editing

### Author ORCIDs
Melissa J Sharpe (iD) http://orcid.org/0000-0002-5375-2076
Geoffrey Schoenbaum (iD) http://orcid.org/0000-0001-8180-0701

### Ethics
Animal experimentation: This study was performed in strict accordance with the recommendations in the Guide for the Care and Use of Laboratory Animals of the National Institutes of Health. All of the animals were handled according to approved institutional animal care and use committee (IACUC) protocols (#15-CNRB-108) of the NIDA-IRP. The protocol was approved by the ACUC at the IRP (Permit Number: A4149-01). Every effort was made to minimize suffering.

### Decision letter and Author response
Decision letter https://doi.org/10.7554/eLife.28362.005
Author response https://doi.org/10.7554/eLife.28362.006

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
