## [Decision Letter]

Thank you for submitting your article "Preconditioned cues have no value" for consideration by *eLife*. Your article has been reviewed by two peer reviewers, and the evaluation has been overseen by Timothy Behrens as the Senior Editor and Reviewing Editor. The following individual involved in review of your submission has agreed to reveal their identity: Nathaniel D Daw (Reviewer #1).

The reviewers have discussed the reviews with one another and the Reviewing Editor has drafted this decision to help you prepare a revised submission.

In Saccada et al., upon which this paper adds, the authors showed that VTA neurons signaled fired for cues that had never been reinforced but where value could be inferred via a model based mechanism. This suggested that the dopamine system can access model-based (or inferred) values. This is an important result but one potential caveat to this result that was raised by during an insight piece written in *eLife* is that there may be some mediated transfer of value onto these "sensory preconditioned" cues. That is inferential processes may happen offline (for example during sleep) transferring value from the cue that has experienced reward onto the associated cue that has not.

In this paper, the authors attempt to address this issue directly by showing that rats will not work for these sensory preconditioned cues but will work for conditioned ones or even for secondary conditioned ones. Combining this with their earlier results (Sadacca et al), they suggest that this rules out a mediated learning explanation of their sensory preconditioning paradigm, implies that sensory preconditioning depends essentially on model-based inferences (and thus that the dopamine activity seen to preconditioned cues in the previous experiment is associated with model-based rather than model-free learning). It is a very clean experiment and a very interesting result, particularly in combination with the previous study. It is an ideal exemplar of the research advance format. There are however some major concerns that need addressing (one is particularly important).

Essential revisions:

I have left the reviews intact below because I know you like them that way.

In discussions, myself and the two reviewers agreed that you need to address the statistical issue raised by both reviewers (most clearly expressed in Nathaniel's review). In all of our views, the story relies critically on a demonstration that value can be inferred on the cue A in the SPC paradigm and we do not agree that your statistics test for this. I think there is no getting around this point. You need to show that A vs. C is significant for magazine entry. Here is the point in question most clearly enunciated:

Copied from R1 below:

“The analysis in Figure 1 and Figure 2 (the core sensory preconditioning and SOC probes) seems to be based on an ANOVA with a factor of reinforced (AB) vs. non (CD) and a factor of first (AC) vs. second (BD) stimulus. This means that the key probe test whether there is actually a preconditioning or SOC effect is based on there being a reinforcer effect AB > BC but no significant stimulus effect AC ~= BD.

I think this is not the right test for preconditioning or SOC. The simple t test A>C, comparing the stimulus to its appropriately matched control, seems like the obvious and appropriate test for these phenomena. Comparing AB vs. CD, as the positive part of the ANOVA does, inappropriately "credits" responding to B vs. D toward a putative preconditioning effect, and it is only by affirming the null hypothesis in comparing AC to BD that this analysis "rules out" the possibility that the AB>CD main effect is driven by B>D rather than A>C. But, of course, this is a fallacy.”

You will also see that there are various issues to do with the interpretation of the study. They pertain principally to whether conditioned reinforcement is a good test of model free value, why model-based value should not support conditioned reinforcement, and in particular why not, if we know that model-based value leads to dopaminergic activity which mediates conditioned reinforcement.

The queries are remarkably consistent between the two reviewers (and are shared by me), so should be taken seriously in the revised text.

Here are the reviews verbatim.

*Reviewer #1:*

This is a super interesting and clean study, which shows a dissociation between sensory preconditioning and conditioned inhibition in terms of whether the resulting CS associations support conditioned reinforcement. Taken (among other things) in light of the authors' previous *eLife* report, this supports a dissociation in the associative structures learned, e.g. that sensory preconditioning is based on stimulus-stimulus or model-based association, and deepens the mystery of the original study's finding of dopamine responses related to the sensory preconditioned cues.

A few suggestions:

- The analysis in Figure 1 and Figure 2 (the core sensory preconditioning and SOC probes) seems to be based on an ANOVA with a factor of reinforced (AB) vs. non (CD) and a factor of first (AC) vs. second (BD) stimulus. This means that the key probe test whether there is actually a preconditioning or SOC effect is based on there being a reinforcer effect AB > BC but no significant stimulus effect AC ~= BD.

I think this is not the right test for preconditioning or SOC. The simple t test A>C, comparing the stimulus to its appropriately matched control, seems like the obvious and appropriate test for these phenomena. Comparing AB vs. CD, as the positive part of the ANOVA does, inappropriately "credits" responding to B vs. D toward a putative preconditioning effect, and it is only by affirming the null hypothesis in comparing AC to BD that this analysis "rules out" the possibility that the AB>CD main effect is driven by B>D rather than A>C. But of course this is a fallacy.

- Without taking anything away from the richness of the result, I feel like the interpretation is a little overly broad. First, I don't think it's appropriate or necessary to draw conclusions about sensory preconditioning in general. Shohamy and Wimmer (2012) (which the authors should absolutely cite as the real evidentiary source for the idea mentioned by Doll and Daw 2016) show good evidence for mediated conditioning driving sensory preconditioning in humans; there is also a literature attributing closely related acquired equivalence effects in rodents in these terms (e.g. Ward Robinson and Hall 1999) though without much direct evidence for the mechanism.

The important thing about the current results is that they show this dissociation from SOC using a common set of procedures, and that these are the same ones previously used to identify dopaminergic correlates. So what's really important is that this demonstrates dissociations between associative value measured several different ways (preconditioning, SOC, pavlovian CRs, dopamine). But given that there is evidence that in other circumstances, mediated conditioning does occur, and that these sorts of preconditioning effects are themselves variable as to whether they even occur at all (e.g. see cites in Ward Robinson paper), we clearly don't understand the factors that govern to what extent mediated conditioning vs. other factors contributes in different circumstances. I think the conclusions (including title, Abstract) should be qualified more.

- Relatedly, the basic framing and interpretation seems to rest on assumption that conditioned reinforcement works exclusively via transmitting model-free value, i.e. that conditioned reinforcement is an unambiguous test of (model-free) "value". Although this might be true, I don't see why this would be expected to be the case. In principle, a stimulus with model-based associations to reward could serve as the incentive for goal-directed behavior, and this might support lever pressing for much the same reason it supports food cup responding no? The Parkinson result doesn't seem to rule this out.

Moreover, it's not even clear that the authors wouldn't expect CRF to work for preconditioned cues, to the extent I understand their view on dopamine. I think the idea is that preconditioned cues can activate dopamine, and also that dopamine can reinforce lever pressing (this is the usual story of how model-free CRF works, supported by data from Everitt), so it's not clear why it doesn't have this effect here. I don't think this in any way cuts against the interest of the paper, but I do think the finding that preconditioning supports dopamine responding, food cup responding, but not conditioned reinforcement, doesn't have a particularly straightforward interpretation in terms of dopamine's relationship to model-based vs model-free learning and it's not entirely clear what the authors are getting at with the last sentence of the Abstract.

*Reviewer #2:*

In this paper, the authors suggest using rats that sensory preconditioned cues are not capable of supporting conditioned reinforcement (whereas, for instance, in an otherwise similar paradigm, secondary conditioned cues are). Combining this with their earlier results (Sadacca et al), they suggest that this rules out a mediated learning explanation of their sensory preconditioning paradigm, implies that sensory preconditioning depends essentially on model-based inferences (and thus that the dopamine activity seen to preconditioned cues in the previous experiment is associated with model-based rather than model-free learning). I think that the results are very interesting - and will be an important contributor to the literature.

I have some questions about the statistics; but am mostly rather puzzledat the interpretation. The trouble is that the key conclusions of thepaper lies on the interpretation of conditioned reinforcement – which isitself far from completely straightforward. In particular, the paperdoes not really explain either why model-based learning cannot/shouldnot support conditioned reinforcement, nor why the dopamine activitythat Sadacca et al. would predict would be inspired by the sensorypreconditioned cues, would not support operant responding here, whenoptogenetically stimulated dopamine responses (for instance)can. Further, the conclusion that no value is attributed to thepre-conditioned cues seems a bit beyond the paper – value is not definedsolely in terms of conditioned reinforcement.

- From Figure 1, it looks superficially as if cue C (which has no reasonto inspire magazine entries) does so more than cue D (which also hasno reason to do so). Is this difference statistically significant byitself? If so, then perhaps something about the prediction that itsupports is important – and this could underpin part of the magazineresponding to A. The statistical test that is done (A&C vs. B&D) doesnot quite tell us the answer to that.

- Although it is an unfair between-subject comparison, it is notablethat the conditioned responding to cue A in Figure 2 does not looksignificantly different to that to cue A in Figure 1. Of course, thekey comparison is with other cues – but it does make one wonder aboutthe relative strengths of the effects (the magazine entries in 2D arealso weaker than those in 1D).

- Why wasn't exactly the same paradigm used as in Sadacca et al. By notdoing that, generalization is obviously weakened.

- Did you score for sign tracking vs. goal tracking (are any of the cuessufficiently approachable to support sign tracking?). This is quiteimportant given results about differential dopaminergic effects insign vs. goal trackers.

---

## [Author Response]

Essential revisions:I have left the reviews intact below because I know you like them that way.In discussions, myself and the two reviewers agreed that you need to address the statistical issue raised by both reviewers (most clearly expressed in Nathaniel's review). In all of our views, the story relies critically on a demonstration that value can be inferred on the cue A in the SPC paradigm and we do not agree that your statistics test for this. I think there is no getting around this point. You need to show that A vs. C is significant for magazine entry. Here is the point in question most clearly enunciated:

We appreciate the desire to see a significant difference between the A vs. C comparison in the SPC probe test, given that we failed to find an A vs. C difference in the conditioned reinforcement. An analysis on the original data yielded a difference only at p=0.053. It is worth noting that the probe test is conducted after the cues are presented a number of times without reward in the course of running the conditioned reinforcement, so some weakness is to be expected as these cues have extinguished somewhat prior to these tests. Nevertheless, to satisfy the reviewers on this point, we ran an extra set of rats using identical procedures to our original study, which allowed us to 1) replicate our finding that preconditioned cues do not promote conditioned reinforcement, and 2) pull out the significant difference between A and C in the probe tests after the conditioned reinforcement tests. We have now added these rats to the original data and updated the results and figures.

You will also see that there are various issues to do with the interpretation of the study. They pertain principally to whether conditioned reinforcement is a good test of model free value, why model-based value should not support conditioned reinforcement, and in particular why not, if we know that model-based value leads to dopaminergic activity which mediates conditioned reinforcement.

We have tried to respond below to the specific queries regarding this point. But we wanted to lay out our general thinking here. Basically, we think this is a terrific set of questions and thinking about it caused us to entirely rewrite the Introduction and Discussion to be clearer as to what we think our data mean. Our main conclusion has not changed, but we hope the new text makes the significance of these data much clearer.

In essence, we agree there is no perfect test of value. We chose to use conditioned reinforcement to ask whether the preconditioned cues have value, because of the general idea that this procedure assesses the subject’s willingness to work to obtain a cue, independent of what the cue predicts. Normally conditioned reinforcement supported by a first-order cue is insensitive to devaluation of the predicted reward (Parkinson, Roberts et al. 2005, Burke, Franz et al. 2007, Burke, Franz et al. 2008), thus a form of value that is presumably model-free or cached in the cue is sufficient to support conditioned reinforcement. That cached value is sufficient is very important – because of this we think that our failure to see any evidence of conditioned reinforcement for the preconditioned cue means, at a minimum, that it does not have such cached or model free value (in our procedure in rats).

This is the essential finding of the study. We think it is hard to argue with really. If preconditioned cues had cached value, then the rats would press the lever to get them. The rats clearly do not do this. This result by itself if meaningful because it rules out the simplest proposal that dopamine neurons fire to preconditioned cues in this task because they have accrued model-free value.

Less clear but perhaps more intriguing is what our failure to find conditioned reinforcement means for proposals that value can also be model-based or inferred. In saying this, we are referring to the idea that because A predicts B and B predicts food, when the rat is presented with A, it evokes a representation of the food and thereby triggers some “value” representation in a model-based manner – presumably the value cached in the sensory features of that food. The model-free value of the food. We believe that this sort of representation exists. Indeed it is how we have in the past thought of the behaviour of the rats in the probe test in preconditioning (Jones, Esber etal. 2012). Further, we think this sort of representation is also sufficient to support conditioned reinforcement, since we know that in some cases, instrumental responding for a cue directly paired with reward is sensitive to devaluation (Burke, Franz et al. 2007, Burke, Franz et al. 2008).

The resolution to this conundrum is not clear. We would speculate that the difference lies in the fact that a preconditioned cue is not the same as a cue that is directly paired with a reward (or with a reward predicting cue). Because it is never directly paired with anything that has any value of its own, it may trigger representations of the downstream entities (the second cue or the food reward) that are dissociable from any sort of cached or model-free value that those entities possess. Perhaps when the rat sees A it understands the associative relationship to the reward but does not, by default, access its associated value. If one accepts this line of reasoning, then the dopamine response to the preconditioned cue may not be about value at all – accessed either directly or indirectly – rather it is about the unexpected information provided by the cue. This is obviously consistent with proposals by Ethan Bromberg-Martin and our own in press data showing that dopamine neurons signal prediction errors for valueless changes in sensory information (Bromberg-Martin and Hikosaka 2009, Takahashi, Batchelor et al. 2017).

We would hasten to add that our explanation for why we don’t see conditioned reinforcement for the putative model-based value of the preconditioned cue is highly speculative. So, we bring this up only at the end of the Discussion, and we have tried to separate it from the more straightforward point that the cue cannot have any intrinsic cached or model-based value, and what this means for our prior results.

The queries are remarkably consistent between the two reviewers (and are shared by me), so should be taken seriously in the revised text.Here are the reviews verbatim.Reviewer #1:[…] A few suggestions:- The analysis in Figure 1 and Figure 2 (the core sensory preconditioning and SOC probes) seems to be based on an ANOVA with a factor of reinforced (AB) vs. non (CD) and a factor of first (AC) vs. second (BD) stimulus. This means that the key probe test whether there is actually a preconditioning or SOC effect is based on there being a reinforcer effect AB > BC but no significant stimulus effect AC ~= BD.I think this is not the right test for preconditioning or SOC. The simple t test A>C, comparing the stimulus to its appropriately matched control, seems like the obvious and appropriate test for these phenomena. Comparing AB vs. CD, as the positive part of the ANOVA does, inappropriately "credits" responding to B vs. D toward a putative preconditioning effect, and it is only by affirming the null hypothesis in comparing AC to BD that this analysis "rules out" the possibility that the AB>CD main effect is driven by B>D rather than A>C. But of course this is a fallacy.

As described in our general response, we appreciate the reviewer’s desire to see a direct comparison of Pavlovian responding to the preconditioned cues, given our failure to see conditioned reinforcement for those cues. Unfortunately, in our original experiment, this difference did not quite reach significance in isolation (p = 0.053). To address this, we have now run an additional cohort of rats. These rats behaved similarly to the prior group – they failed to show conditioned reinforcement for A while showing a subsequent difference in Pavlovian responding to A versus C. When combined with the prior subjects, the difference reached significance. In the revised manuscript, we have combined them in the results and figure.

- Without taking anything away from the richness of the result, I feel like the interpretation is a little overly broad. First, I don't think it's appropriate or necessary to draw conclusions about sensory preconditioning in general. Shohamy and Wimmer (2012) (which the authors should absolutely cite as the real evidentiary source for the idea mentioned by Doll and Daw 2016) show good evidence for mediated conditioning driving sensory preconditioning in humans; there is also a literature attributing closely related acquired equivalence effects in rodents in these terms (e.g. Ward Robinson and Hall 1999) though without much direct evidence for the mechanism.The important thing about the current results is that they show this dissociation from SOC using a common set of procedures, and that these are the same ones previously used to identify dopaminergic correlates. So what's really important is that this demonstrates dissociations between associative value measured several different ways (preconditioning, SOC, pavlovian CRs, dopamine). But given that there is evidence that in other circumstances, mediated conditioning does occur, and that these sorts of preconditioning effects are themselves variable as to whether they even occur at all (e.g. see cites in Ward Robinson paper), we clearly don't understand the factors that govern to what extent mediated conditioning vs. other factors contributes in different circumstances. I think the conclusions (including title, Abstract) should be qualified more.

We apologize if our conclusions were too broad, and we have tried in our revision to emphasise that our results are most relevant for our procedure. Most importantly, we did not mean to imply that our data rule out a role for mediated conditioning in sensory preconditioning. Rather, they suggest that if mediated conditioning does occur, as for example suggested by Wimmer and Shohamy’s data (Wimmer and Shohamy 2012), it likely does not allow the preconditioned cue to access or accrue model-‐‑free value (again at least in our procedure in rats). Instead the preconditioned cues may be activated during conditioning to acquire the ability to activate representations of either the directly conditioned cue and/or the sensory specific properties of the food reward, independent of any value stored in them. This would qualify as mediated learning and yet still not support conditioned reinforcement. Or, alternatively, our results may not apply to other settings. We have added text to the Discussion to address these points.

- Relatedly, the basic framing and interpretation seems to rest on assumption that conditioned reinforcement works exclusively via transmitting model-free value, i.e. that conditioned reinforcement is an unambiguous test of (model-free) "value". Although this might be true, I don't see why this would be expected to be the case. In principle, a stimulus with model-based associations to reward could serve as the incentive for goal-directed behavior, and this might support lever pressing for much the same reason it supports food cup responding no? The Parkinson result doesn't seem to rule this out.Moreover, it's not even clear that the authors wouldn't expect CRF to work for preconditioned cues, to the extent I understand their view on dopamine. I think the idea is that preconditioned cues can activate dopamine, and also that dopamine can reinforce lever pressing (this is the usual story of how model-free CRF works, supported by data from Everitt), so it's not clear why it doesn't have this effect here. I don't think this in any way cuts against the interest of the paper, but I do think the finding that preconditioning supports dopamine responding, food cup responding, but not conditioned reinforcement, doesn't have a particularly straightforward interpretation in terms of dopamine's relationship to model-based vs model-free learning and it's not entirely clear what the authors are getting at with the last sentence of the Abstract.

Just to recap our views here (see also our remarks to the essential revisions above), we believe that model-free value is sufficient to support conditioned reinforcement, since it has been shown that conditioned reinforcement for directly conditioned cues is normally insensitive to reward devaluation (Parkinson, Roberts et al. 2005). As a result, our failure to find conditioned reinforcement for the preconditioned cues provides strong evidence that these cues (in our procedure in rats) differ from normal cues in that they do not possess such model-free value.

A more difficult question is whether they have the ability to link to other representations that access such value – such as the other cue or the food itself. Certainly we believe this is the case for a cue directly paired with food reward – that is while it has a cached value representation that is sufficient to support conditioned reinforcement, we would agree that it likely also has value through its direct association to the food (essentially the value cached in the sensory properties of the food) that can also support instrumental responding. This is something we have explored in other studies (Burke, Franz et al. 2007, Burke, Franz et al. 2008).

However here we are testing a cue that has never been paired with food or anything else that has value. So we believe our failure to see any evidence of conditioned reinforcement indicates that the associative representations triggered by the preconditioned cue do not, by default, dredge up any remote value representations via the associative model. In this, the preconditioned cues could be fundamentally different from any directly conditioned cue. They are purely informational. As we noted above, this has implications about what dopamine neurons may be signalling when these cues are encountered that go beyond whether it is a model-free or model-based value signal.

We have tried to make clear in the Discussion, first, that we think our data rule out a simple model-free value explanation of the firing of the dopamine neurons, and then second, that we would speculate the results pose additional problems for explaining the dopamine response as a value signal at all. Obviously the second point is highly speculative, so we have tried to indicate that. And we can remove or curtail it. But we think the reviewer has identified a very important implication of the data that should be discussed.

Reviewer #2:In this paper, the authors suggest using rats that sensory preconditioned cues are not capable of supporting conditioned reinforcement (whereas, for instance, in an otherwise similar paradigm, secondary conditioned cues are). Combining this with their earlier results (Sadacca et al), they suggest that this rules out a mediated learning explanation of their sensory preconditioning paradigm, implies that sensory preconditioning depends essentially on model-based inferences (and thus that the dopamine activity seen to preconditioned cues in the previous experiment is associated with model-based rather than model-free learning). I think that the results are very interesting – and will be an important contributor to the literature.I have some questions about the statistics; but am mostly rather puzzled at the interpretation. The trouble is that the key conclusions of the paper lies on the interpretation of conditioned reinforcement – which is itself far from completely straightforward. In particular, the paper does not really explain either why model-based learning cannot/should not support conditioned reinforcement, nor why the dopamine activity that Sadacca et al. would predict would be inspired by the sensory preconditioned cues, would not support operant responding here, when optogenetically stimulated dopamine responses (for instance) can. Further, the conclusion that no value is attributed to the pre-conditioned cues seems a bit beyond the paper – value is not defined solely in terms of conditioned reinforcement.

We apologize for the confusion. Please see our responses above to the general comments and reviewer 1. Briefly we agree that model-free value is likely not the only type of value to support conditioned reinforcement for a cue directly paired with reward. However based on data showing that such responding is insensitive to reward devaluation (Parkinson, Roberts et al. 2005), value cached in the cue is clearly sufficient. So we believe our failure to see any evidence of conditioned reinforcement for a preconditioned cue means this cue does not possess its own (model‑free, cached) value. This has obvious implications for why dopamine neurons fire to this cue in our hands.

However, we also agree that, for a reward paired cue, there might also be a model-based value that could support responding. Indeed we have made such a suggestion (Burke, Franz et al. 2007, Burke, Franz et al. 2008). Yet a preconditioned cue has never been paired with anything of value. In this way, it is different from cues that we tested in these studies (Burke, Franz et al. 2007, Burke, Franz et al. 2008). We would speculate that our failure to see any conditioned reinforcement for the preconditioned cue means that it is not accessing cached value, even indirectly in a model-based manner. This obviously has further implications for why dopamine neurons fire to this cue. We have added to the Discussion to make these points.

- From Figure 1, it looks superficially as if cue C (which has no reason to inspire magazine entries) does so more than cue D (which also has no reason to do so). Is this difference statistically significant by itself? If so, then perhaps something about the prediction that it supports is important – and this could underpin part of the magazine responding to A. The statistical test that is done (A&C vs. B&D) does not quite tell us the answer to that.

We appreciate the reviewer’s concern. Indeed there was a main effect of cue type (AC vs. BD; *F*_(1,21)_=9.39, *p*<0.05); although critically there was no interaction (*F*<1). We believe this difference in responding to the preconditioned versus the directly conditioned cues occurs because the AC extinction is run before the BD extinction. We did this on purpose, since the AC comparison is so critical (as indicated by reviewer 1’s comments), and rats normally reduce even baseline responding quickly once we start extinction testing. We have added text to the results to explain this.

Most importantly, this general difference in responding simply highlights that C is the control for A. It is chosen to be similar in both modality and salience – for example, in both designs, we counterbalance the identity of A and C (white noise or click) and B and D (tone or siren) separately – and C is treated as similarly as possible to A in terms of the number of times and conditions under which it is presented. For this reason, it is the appropriate comparison for A. Comparing to D ignores all of these differences, any of which might lead to differences in responding for trivial reasons.

- Although it is an unfair between-subject comparison, it is notable that the conditioned responding to cue A in Figure 2 does not look significantly different to that to cue A in Figure 1. Of course, the key comparison is with other cues – but it does make one wonder about the relative strengths of the effects (the magazine entries in 2D are also weaker than those in 1D).

As the reviewer notes, this is a comparison made between subjects and across cues that have had different training and treatment. Indeed rates of responding are slightly lower across the board in the SOC experiment. This may reflect training differences across subjects in the two studies. For example, in the SOC experiment, rats first receive conditioning, then second-order conditioning. This effectively constitutes an extinction session before they go into the conditioned reinforcement tests. Thus the rats may show lower responding since they have had more experience with the cues and no reward (and all the more impressive that they show conditioned reinforcement for the second-order cue). These potential differences are a key reason we used a within-subjects design.

- Why wasn't exactly the same paradigm used as in Sadacca et al. By notdoing that, generalization is obviously weakened.

The procedure is substantially the same. The only significant deviation we can find is the expansion of conditioning by two sessions, versus Sadacca et al. (2016). This is a fairly trivial change and was not purposeful but rather one of the many ways we run this effect in the lab (e.g. we recently used this specific version of the task to implicate a causal role for dopamine in sensory preconditioning (Sharpe, Chang et al. 2017). It is also the same as the conditioning used in the SOC study here, so we don't think the longer conditioning is likely to have caused the difference in conditioned reinforcement we observed for SPC vs. SOC cues.

- Did you score for sign tracking vs. goal tracking (are any of the cues sufficiently approachable to support sign tracking?). This is quite important given results about differential dopaminergic effects in sign vs. goal trackers.

This is an interesting point raised by the reviewer. However, unfortunately we cannot look at those behaviours. The levers were not inserted as cues, which is typically how sign tracking is done; our levers were not retractable and remained accessible for the entire conditioned reinforcement session in order to encourage responding, and of course we did not have a food port either. So, it is not possible to measure sign or goal tracking.